# Drilling-Induced Damages in Hybrid Carbon and Glass Fiber-Reinforced Composite Laminate and Optimized Drilling Parameters

Elango Natarajan [1,2,*], Kalaimani Markandan [1], Santhosh Mozhuguan Sekar [3], Kaviarasan Varadaraju [4], Saravanakumar Nesappan [2], Anto Dilip Albert Selvaraj [2], Wei Hong Lim [1] and Gérald Franz [5,*]

1. Faculty of Engineering, Technology and Built Environment, UCSI University, Kuala Lumpur 56000, Malaysia
2. Department of Mechanical Engineering, PSG Institute of Technology and Applied Research, Coimbatore 641062, India
3. Selvam Composite Materials Research Laboratory, Department of Mechanical Engineering, Selvam College of Technology, Namakkal 637003, India
4. Department of Mechanical Engineering, Sona College of Technology, Salem 636005, India
5. Laboratoire des Technologies Innovantes, UR UPJV 3899, Avenue des Facultés, Le Bailly, 80025 Amiens, France
* Correspondence: elango@ucsiuniversity.edu.my (E.N.); gerald.franz@u-picardie.fr (G.F.)

**Abstract:** Hybrid carbon and glass fiber-reinforced composites have attracted significant research interest for primary load-bearing structural components in the field of aviation manufacturing owing to their low weight and high strength to weight ratio. However, the anisotropic and heterogenic nature of carbon and/or glass fiber-reinforced composite prevents high machining quality due to the directionality effect of fibers in the polymer matrix. As such, this study investigates the effect of drilling process for hybrid fiber-reinforced composite and reports optimal drilling parameters to improve the drill quality. Experimental studies indicate that an increased point angle (i.e., from 80° to 120°) resulted in low delamination upon entry due to reduced thrust force, which in turn produces better surface finish with minimal tool wear. The optimal feed rate (0.2 mm/min) ensures lower delamination at entry, since higher feed rates can increase the thrust force due to elevation in the shear area or raise the self-generated feed angle, which in turn reduces the effective clearance angle. To this end, drilling parameters were optimized using Dandelion optimizer (DO)—a cutting-edge metaheuristic search algorithm (MSA). We report the excellent consistency of DO to solve the proposed drilling optimization problem while achieving promising results as ascertained by the small standard deviation values.

**Keywords:** dandelion optimizer; carbon fiber; S-glass; hybrid; drilling; laminate

## 1. Introduction

Drilling is a common machining operation which involves material-removal for the applications such as screws or rivets in the assembly of parts. As such, high quality of the holes is an essential factor to ascertain the accuracy of assembly with other structural parts upon drilling [1,2]. However, few studies have consistently highlighted the various damages during drilling operation caused by the inhomogeneity and anisotropy of composites such as fiber pull-outs, fiber–matrix debonding or thermal damage [3,4]. Delamination is a common damage that occurs upon drilling of composite holes, which results in damage bonder surrounding the drilled hole. Delamination results in poor assembly tolerance and hampered structural integrity, which in turn affects the long-term performance of materials under fatigue loads. In a study by Seif et al. [5], it was reported that 60% of part rejection in aircraft was attributed to damage due to delamination.

Delamination can be caused by peel-up or push-down mechanisms when the thrust force exerted by the drill exceeds the interlaminar fracture toughness of layers [4,6]. Peel-up

is the result of the upper layer of materials that are pushed through the drilled faces, while the push-down defect occurs due to the indentation effect by the quasi-stationary drill chisel edge that acts on uncut layers of laminate. In most cases, peel-up delamination can be prevented by using low feed rates [7], while push-down delamination can be prevented by using a support plate under the material to be drilled [8]. Analogously, the value of thrust force which affects the delamination of layers is strongly dependent on various factors such as the property of the material to be drilled (i.e., strength, stiffness), drill geometry or machining parameters such as spindle speed, feed rate or the use of coolant [9–11]. In other studies, it has been highlighted that drilling speed plays a vital role in reducing the radial thrust force, narrowing the specific energy map and thereby improving the machinability of various composites [12,13].

Today, carbon fiber-reinforced polymer (CFRP) composites are commonly used as a major material for primary load-bearing structural components in the field of aviation manufacturing owing to their low weight and high strength to weight ratio. For instance, CFRP has been utilized for various load-bearing structures in large civil aircrafts such as B737, A380, A350 and A400M [12,14]. In a more recent study, the suitability of carbon fiber (CF) and glass fiber (GF)-reinforced epoxy for a submarine hull subjected to hydrostatic pressure was investigated [15]. Most structures obtained from CFRP generally require post-processing machining operations such as drilling to meet the geometric dimension, shape accuracy and surface quality of the final structure. However, the anisotropic and heterogenic nature of fiber-reinforced plastics leads to machining process that differs from most metallic materials attributed to the directionality effect of fibers in the polymer matrix. For instance, it was reported that the interlayer bonding strength is 5–20% of the tensile strength along the fiber direction, which may result in interlayer delamination during machining process [12]. In addition, the prominent anisotropy and heterogeneity leads to a poor machinability of layers.

In a study by Xu et al., it was reported that CF-reinforced polyetheretherketone (PEEK) exhibited poorer machinability (i.e., higher drilling force, higher cutting temperature, larger delamination and excessive tool wear) compared to CF-reinforced polyimide (PI) composites [16]. It was highlighted that CF-PEEK composites exhibit ductile behavior to a certain extent and undergo plastic deformation during the drilling process, since machining temperature exceeds the glass transition temperature of PEEK. On the other hand, the CF-PI composites exhibit a brittle fracture behavior during the drilling process. In two-part comprehensive studies, Gemi et al. investigated for the time the drilling performance of a hybrid composite consisting of glass and carbon fiber in various stacking sequences [17,18]. The authors reported that stacking sequence significantly affects the mechanical properties and thrust force generated during drilling; a carbon layer stacked as the last ply on the inner surface resulted in less push-out delamination, since the glass layer was more sensitive to delamination than the carbon layer. The composite laminates were less prone to delamination at a lower feed rate and higher cutting speed of the composites.

In the present study, we first investigate the effect of drilling parameters such as drill bit spindle speed, feed rate and point angle on the delamination upon entry of CFRP laminates. Later, the effect of drilling parameters was optimized using Dandelion optimizer (DO) to attain minimal delamination of the composite laminate. Based on our optimization findings, we report an optimal spindle speed of 2400 rpm, feed rate of 0.3 mm/min and point angle of 89° to ensure minimal predicted delamination of 0.8644 (bottom side), which agreed consistently with experimental findings. While previous studies have consistently reported on the effects of drilling parameters on delamination of composite materials [19–21], to the best of authors' knowledge, this is the first study that has attempted to apply DO for searching the best combinations of machining parameters for drilling optimization problem. To this end, Section 2 of the present study reports preparation of the composite laminate while Section 3 reports the mechanical properties, morphological analysis as well as the delamination analysis of the laminate. Optimization of drilling parameters using DO

algorithm and performance comparison of all nature-inspired algorithms in solving the proposed drilling optimization problem is presented in Section 4.

## 2. Materials and Methods

### 2.1. Preparation of Composite Material

The preparation of laminated composites used in the present study is similar to our previous study [15]. In short, epoxy resin (LY 556) with viscosity of 8000–12,000 cPS and a hardener (HY 951) with viscosity of 16,000–18,000 cPS was mixed using a mechanical stirrer at a ratio of 10:1. GF and CF woven mats with densities of 1.95 g/cm$^3$ and 2.40 g/cm$^3$, respectively, were purchased from a local supplier to be used as reinforcement in the epoxy matrix. Three different laminates were prepared in the present study, which include (i) GF/epoxy laminate (GFRP), (ii) CF/epoxy laminate (CFRP) and (iii) hybrid CF/GF/epoxy laminate (C/GFRP).

A steel plate with dimensions of 455 mm × 300 mm × 5 mm (length × width × thickness) was used as the basin for molding where the plate was covered with a perforated release film made of a thin glass sheet. The fabrication procedure for all laminates was similar. For instance, to prepare the aforementioned laminate (iii), epoxy resin was applied over the thin glass sheet after which the first layer of S-glass mats was placed. For the second layer, epoxy resin was applied over the first layer, and subsequently, a CF mat was placed over it. The stacking process with epoxy resin was repeated until a laminate with a thickness of 3 mm was obtained. The laminate consisted of three layers each of GF mats and CF mats, as illustrated schematically in Figure 1 with approximate volume fractions of 40% and 14%, respectively. Upon completion of the stacking process, the laminates were compression molded for 2 h at 150 bar. The process was completed by removing the laminates from the mold and curing at 55 °C for 24 h.

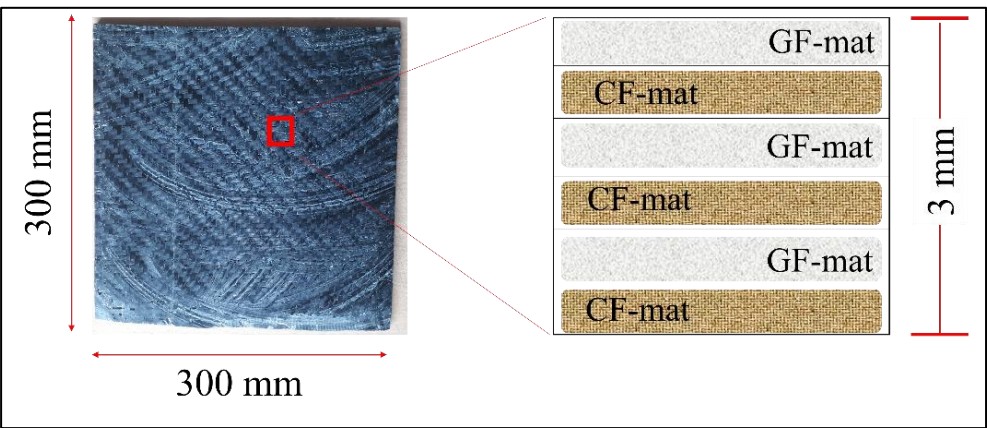

**Figure 1.** Schematic illustration of epoxy laminates reinforced with GF and CF mats with thickness of 3 mm.

### 2.2. Characterizations

Microstructural characterizations were performed using Scanning Electron Microscopy (SEM), JSM-7600F (JEOL Inc., Tokyo, Japan) with an accelerating voltage of 2–5 kV. Prior to microstructural analysis, the surfaces of laminates were coated with ≈4 nm of Pt using a Hitachi S-3000N machine. The 3rd generation Empyrean, Malvern Panalytical multipurpose diffractometer with MultiCore Optics was used for X-ray diffraction (XRD) analysis. Raman spectroscopy analysis was performed using a Renishaw He–Ne laser with 633 nm wavelength coupled to an optical microscope (Leica DM 2500 M). The ability of the laminates to withstand axial stresses was investigated using a universal testing machine (Instron 4855, Norwood, MA, USA) equipped with 100 kN load cell for tensile testing according to ASTM D3039 standards (120 mm × 20 mm × 3mm). The ability of the laminates to withstand transverse loads was evaluated using the 3-point bending testing according to

ASTM-D-790 on samples having dimensions of 100 mm × 20 mm × 5 mm. The energy absorption capability of the laminates was investigated using the Izod impact testing machine according to ASTM D256-04. Unless otherwise stated, all samples were tested three times, and the average results were reported.

## 3. Results and Discussion

### 3.1. Morphology Characterizations

Figure 2A–F show the SEM images at low and high magnifications, respectively, for CFRP, GFRP and C/GFRP. From Figure 2, it can be seen that CF and GF were well dispersed and impregnated into the epoxy matrix. The CF and GF exhibited good interfacial bonding and excellent adhesion with the epoxy matrix. From Figure 2D–F, it can be ascertained that the long CF and GF will improve the load-carrying capability of the composites due to increased transfer length. This in turn ensures effective stress transfer from the epoxy matrix to CF and GF, which is beneficial to enhance the mechanical properties such as the stiffness, tensile and flexural strength of the composites [22]. For example, when load is applied, crack initiation and propagation occur. During crack propagation, when a crack meets the GF or CF, it is arrested and deflected in plane, which is representative of a complex pathway to release stress that enhances the toughness of the epoxy-reinforced polymer composites [23].

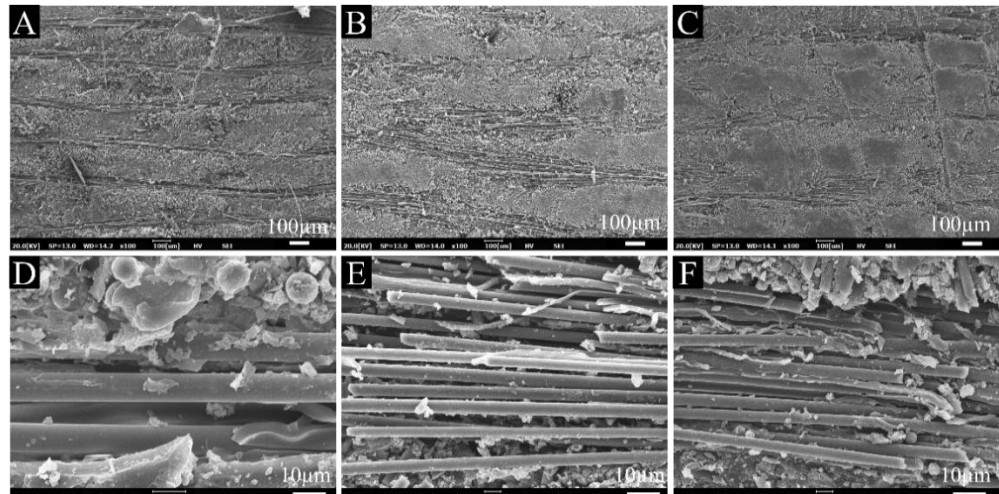

**Figure 2.** SEM images of epoxy composites reinforced with (**A**) CF, (**B**) GF, (**C**) CF/GF, and (**D–F**) magnified images of composites (**A–C**), respectively.

### 3.2. X-ray Diffraction (XRD) Analysis

Figure 3A–C show the XRD analysis of GFRP, CFRP and hybrid C/GFRP, respectively, where the scattering angle (2-theta) corresponds to the intensity of the composites. The XRD analysis in the present study based on the diffraction patterns of materials reveals information regarding the deviation of structure from the ideal structure due to internal stresses and defects. From Figure 3, it can be seen that the composites exhibited visible 2θ peaks at 13.96°, 16.73°, 18.39° and 25.31°. In a study by Wang et al. [24], it was reported that variation in super molecular structures and crystalline states leads to differences in the mechanical properties. For instance, the simplest epoxy is a three-member ring structure which forms 'alpha-epoxy', observed at 2θ = 13.96° and 16.73°, which leads to the high strength and modulus of the epoxy [25]. Specifically, the peak at scattering angle 2θ = 18.38° has been reported to correspond to $C_5H_9NO_2$, which are the organic components of neat epoxy, and it ascertains the cross-linked network between epoxy and hardener and the amorphous nature of neat epoxy [19]. The peak at scattering angle 2θ = 25.31° which is more significant in CFRP and hybrid C/GFRP has been reported in previous studies to correspond to carbon reinforcement [26].

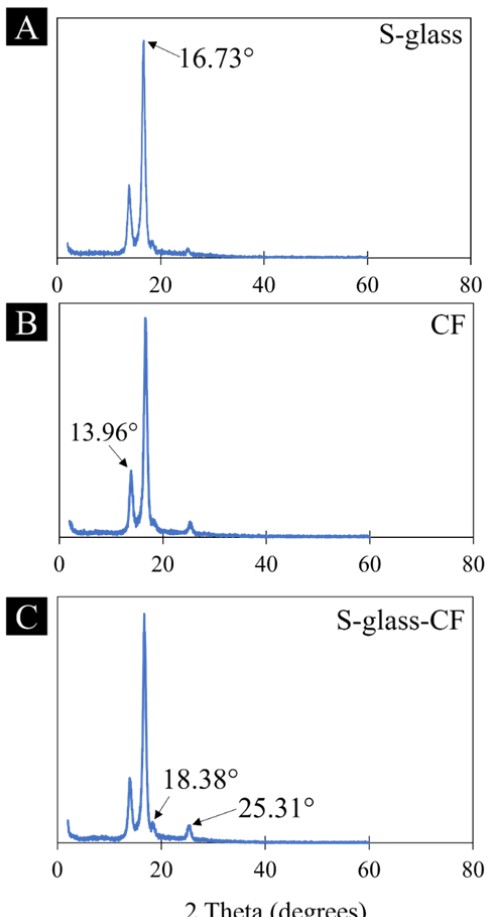

**Figure 3.** XRD analysis of (**A**) GFRP, (**B**) CFRP, and (**C**) C/GRP.

### *3.3. Mechanical Properties of Laminates*

Figure 4 shows the mechanical properties of the epoxy composites reinforced with CF, GF and hybrid CF/GF. From Figure 4A–C, it can be seen that the optimum tensile, flexural and impact properties were obtained for the hybrid C/GFRP. For instance, the elastic modulus of hybrid C/GFRP was improved by 7.48% and 5.79% in tension to CFRP or GFRP. Similarly, the maximum stress during flexural testing for hybrid C/GFRP was 56.3% and 49.7% higher than CFRP and GFRP, respectively. The obtained values for the modulus and stiffness of CFRP, GFRP and C/GFRP agree well with findings reported in the literature [27,28]. On the other hand, the energy absorption of CFRP and GFRP upon low velocity impact was only 8.1 J and 9.1 J, respectively, whereas the hybrid C/GFRP exhibited a higher energy absorption value of 11.3 J. In our previous study, we have highlighted that the hybrid C/GFRP exhibits excellent specific strength compared with steel, Al 7075, magnesium alloy or many common metal alloys [15]. The excellent mechanical properties of all composites in the present study in comparison to the epoxy matrix can be attributed to the excellent interfacial bonding such as interlaminar adhesion and delamination resistance of glass and carbon fibers with epoxy matrix [29,30]. The high energy absorption of the hybrid C/GFRP is also indicative of its tough nature that is suitable for applications requiring post-processing machining operations such as drilling.

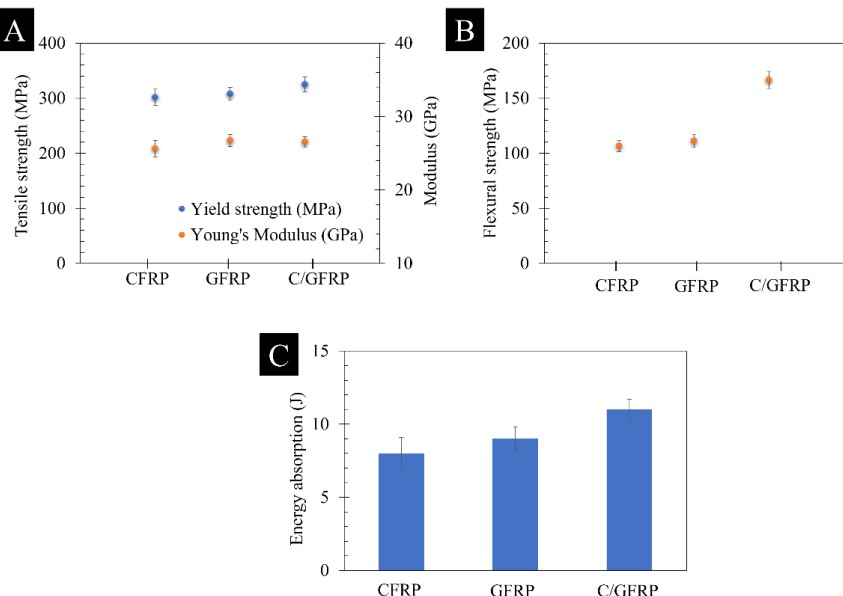

**Figure 4.** Mechanical properties of epoxy reinforced with CF and/or S-glass fiber composites: (**A**) tensile strength and Young's modulus, (**B**) flexural strength, and (**C**) energy absorption of composites upon low-velocity impact.

### 3.4. Raman Analysis

Figure 5 shows the Raman spectrum of hybrid C/GFRP where several significant peaks were identifiable. Among them, the peaks ranging from 2965 to 3079 cm$^{-1}$ can be assigned to vibrations of the methyl (C-H) groups [31], while the peak at 1019 cm$^{-1}$ can represent the epoxy resin backbone vibration [32]. The peak at 1390 cm$^{-1}$ (D-peak) can be assigned to the disordered carbon structure or the polycrystalline graphite related to the boundaries of graphite crystals, while the peak at 1620 cm$^{-1}$ (G-peak) can be assigned to graphite structure [33,34]. Similar to other studies, in the present study, the integrated intensity ratio (Id/Ig) of the carbon fiber which reveals the disorder of carbon fibers was determined to be 0.96 [35]. This indicates that processing of the laminate did not modify the properties of the carbon fiber.

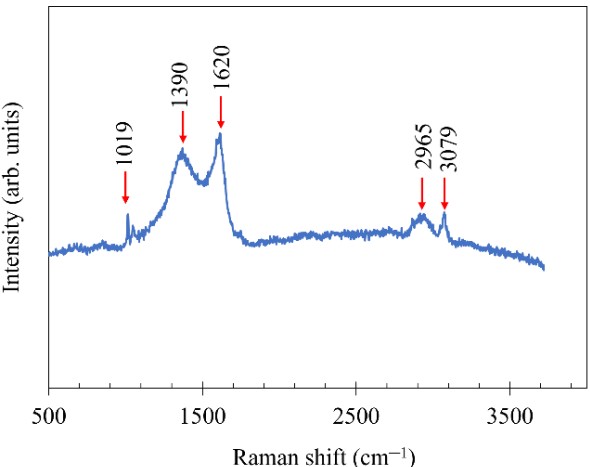

**Figure 5.** Raman spectrum of hybrid C/GFRP.

### 3.5. Drilling and Delamination Analysis

Hybrid fiber composites have gained significant research interest in the past decade. In a study by Chabaud et al. [36], continuous glass fiber and carbon fiber-reinforced polyamide (PA) composites were investigated for potential application in outdoor structural

applications. In another study, Jesthi et al. [37] investigated the mechanical properties of hybrid CF/GF-reinforced polymer composites and reported their suitability for use in marine applications. It is common for the composite materials to be subjected to machining operations such as drilling to remove some parts of the material to enable the application of screws or rivets in the assembly of parts. Thus, it is important to perform drilling and delamination analysis to investigate the effect of machining operation on the fracture mechanics and mechanical properties of the composite material.

When a chisel edge such as a wedge is pushed into the workpiece during the drilling process, the cutting lips begin to cut materials [38]. When the drill advances, some layers of the workpiece tend to be pushed toward the cutting surfaces. The delamination process takes place over the workpiece when the thrust force ($F_z$) exerted by the drill exceeds the interlaminar fracture toughness of the layers. Figure 6 shows the delamination mode during peel-up and push-out as well the geometric parameters for drilling investigation.

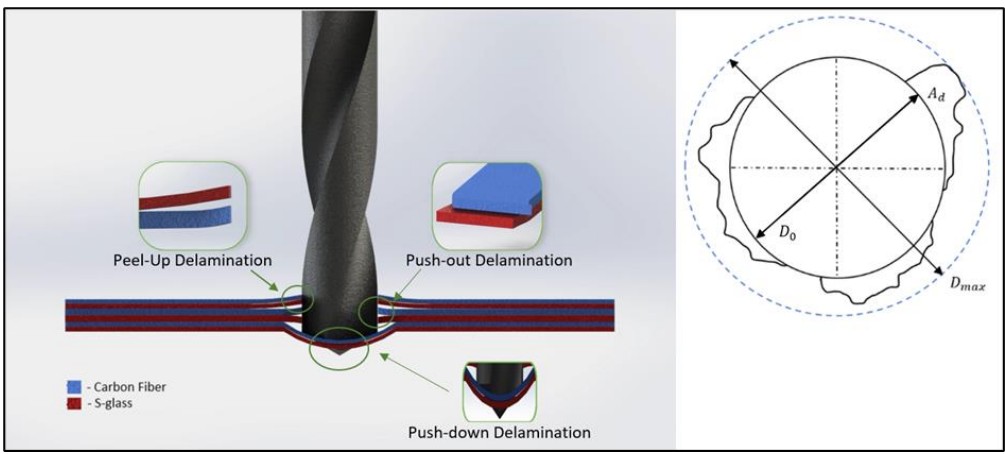

**Figure 6.** Delamination during drilling and schematic illustration of geometric parameters for drilling investigation.

The delamination factor is a measure of the hole-making method, and it reveals the effect of the various drilling parameters. A high delamination factor reflects the low strength of the hole. The delamination factor ($F_d$) is the ratio of the maximum delaminated diameter ($D_{max}$) and nominal diameter of the hole ($D_0$) [39]. It is numerically expressed as:

$$F_d = \frac{D_{max}}{D_0},\tag{1}$$

The tool maker's microscope, acoustic emission test, ultrasound technique, image processing, C-scan, radiography technique, and CT scan may be employed to determine the delamination extension. It is important to capture the surrounding area of the hole for data analysis. The uncut fiber factor (UCFF) can be numerically estimated as [40]:

$$\text{UCFF} = \frac{A_0}{A_H} \text{ or } \frac{A_1}{A_H}\tag{2}$$

where $A_0$ is the area between the hole circle and maximum delamination zone, $A_1$ is the area between the hole circle and the minimum damage zone and $A_H$ is the area of the drill.

### 3.6. Design of Experiments (DoE)

The drill bit spindle speed and feed rate are the most influencing parameters in drilling delamination [41–43]. In the present study, an experimental design was performed with three levels each of spindle speed, feed rate and point angle, as tabulated in Table 1. These levels were selected based on the advice from the material supplier.

**Table 1.** Levels of drilling parameters.

| Parameters | Level 1 (Lower) | Level 2 (Middle) | Level 3 (Maximum) |
|---|---|---|---|
| Spindle speed (rpm) | 1000 | 1700 | 2400 |
| Feed rate (mm/min) | 0.1 | 0.2 | 0.3 |
| Point angle (°) | 80 | 100 | 120 |

Twenty-seven unique experiments were conducted in a vertical milling machine (Deckel-Maho-Gildemeister, DMG VMC 810 Milling, Germany) with Fanuc Controller). All experiments were conducted at room temperature with no coolant used. Hybrid C/GFRP was placed over the bed of the vertical machining center, and an uncoated tungsten carbide (K20) conventional twist drill of 6 mm diameter was fed into the workpiece. After drilling was completed, the delamination was assessed by the flatbed scanning method. The scanned images were imported and analyzed in Image, which is an image processing software. Both the front and back of the drilled hole were assessed as shown in Figure 7.

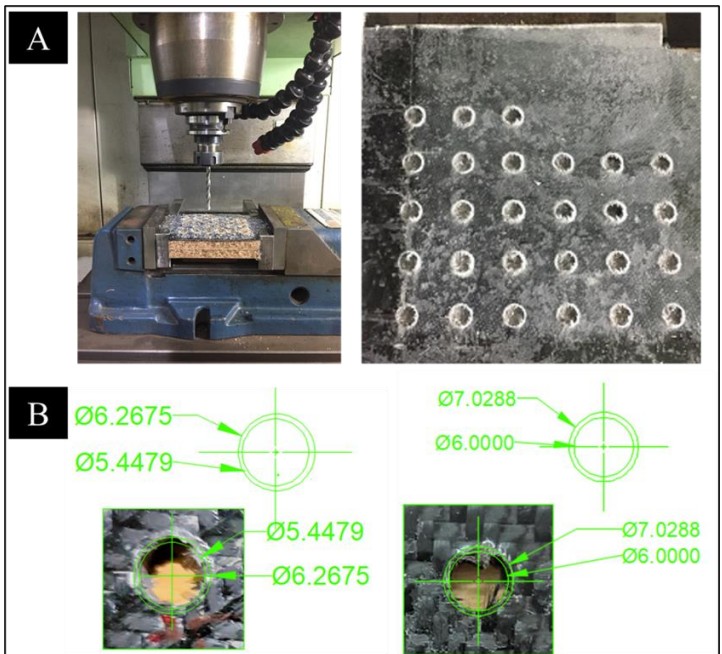

**Figure 7.** (**A**) Drilling operation of samples and images of drilled holes on the surface of the composite material; (**B**) Assessment of diameters D-Max and D-Min at the top and bottom, respectively.

Table 2 summarizes the computed delamination for 27 different combinations of drilling parameters. From Table 2, it can be seen that increased point angle from 80° to 120° generally resulted in lower delamination upon entry. This can be attributed to the reduced thrust force, which in turn also produces a better surface finish with minimal tool wear. Lotfi et al. [44] reported that the reduced point angle increases the resisting moment of forces during drilling. In the present study, it was also noted that a spindle speed of 1000 rpm resulted in lower delamination at entry compared to the higher spindle speeds investigated. This can be attributed to the increased drill vibration at higher speed in particular when hole circularity might deteriorate with depth due to a gradual increase in thermal load during drilling [45]. The feed rate of 0.2 mm/min resulted in lower delamination at entry compared to feed rates of 0.1 to 0.3 mm/min. Higher feed rates can potentially increase the thrust force due to the elevation in shear area. These higher forces in turn generate large bending deformation in the layers underneath the drill bit, and upon delamination, the energy stored due to bending will be released as surface energy [46]. A high feed rate raises the self-generated feed angle, which in turn reduces the effective clearance angle.

**Table 2.** Experimental results indicating delamination at various spindle speeds, feed rates and point angles.

| Experiment | Spindle Speed (rpm) | Feed Rate (mm/min) | Point Angle (°) | $F_d$ (Top Side) | $F_d$ (Bottom Side) |
|---|---|---|---|---|---|
| 1 | 1000 | 0.1 | 80 | 1.18294 | 1.0661 |
| 2 | 1000 | 0.1 | 100 | 1.18375 | 1.0664 |
| 3 | 1000 | 0.1 | 120 | 1.17429 | 1.0741 |
| 4 | 1000 | 0.2 | 80 | 1.17716 | 1.0743 |
| 5 | 1000 | 0.2 | 100 | 1.17635 | 1.0753 |
| 6 | 1000 | 0.2 | 120 | 1.16528 | 1.0836 |
| 7 | 1000 | 0.3 | 80 | 1.18417 | 1.1099 |
| 8 | 1000 | 0.3 | 100 | 1.18175 | 1.1115 |
| 9 | 1000 | 0.3 | 120 | 1.17147 | 1.1441 |
| 10 | 1700 | 0.1 | 80 | 1.18213 | 1.0512 |
| 11 | 1700 | 0.1 | 100 | 1.18697 | 1.0514 |
| 12 | 1700 | 0.1 | 120 | 1.18153 | 1.0590 |
| 13 | 1700 | 0.2 | 80 | 1.17740 | 1.0437 |
| 14 | 1700 | 0.2 | 100 | 1.18062 | 1.0445 |
| 15 | 1700 | 0.2 | 120 | 1.17357 | 1.0527 |
| 16 | 1700 | 0.3 | 80 | 1.18547 | 1.0634 |
| 17 | 1700 | 0.3 | 100 | 1.18707 | 1.0649 |
| 18 | 1700 | 0.3 | 120 | 1.17840 | 1.0737 |
| 19 | 2400 | 0.1 | 80 | 1.17497 | 1.0689 |
| 20 | 2400 | 0.1 | 100 | 1.18383 | 1.0690 |
| 21 | 2400 | 0.1 | 120 | 1.18242 | 1.0764 |
| 22 | 2400 | 0.2 | 80 | 1.17130 | 1.0455 |
| 23 | 2400 | 0.2 | 100 | 1.17854 | 1.0462 |
| 24 | 2400 | 0.2 | 120 | 1.17551 | 1.0543 |
| 25 | 2400 | 0.3 | 80 | 1.18042 | 1.0494 |
| 26 | 2400 | 0.3 | 100 | 1.18605 | 1.0508 |
| 27 | 2400 | 0.3 | 120 | 1.18140 | 1.0595 |

## 4. Optimization of Drilling Parameters for Minimum Delamination

### 4.1. Dandelion Optimizer (DO)

Dandelion optimizer (DO) is a cutting-edge metaheuristic search algorithm (MSA) that was proposed very recently by Zhao et al. [47] for solving continuous optimization problems. The working mechanism of DO is inspired by the propagation process of dandelion seeds to new locations through wind for reproduction purposes when they become mature. Depending on the weather conditions and wind speeds, these dandelion seeds can be spread in different heights and distances. For instance, the weather conditions can determine if the new dandelion can grow in the locations nearby or far away from the parent dandelion by governing the flying ability of the dandelion seeds. Meanwhile, the flight distance of the dandelion seed is influenced by the wind speed. The reproduction of new dandelions via seed propagation can be divided into three stages (i.e., rising stage, descending stage and landing stage) that facilitate the derivation of search operators with unique levels of exploration and exploitation strengths. In the next subsection, the optimization process of DO is first described in terms of its initialization process, which is followed by the mechanisms of balancing the exploration and exploitation strengths based on search operators inspired by the rising, descending and landing stages of dandelion seeds.

#### 4.1.1. Initialization of DO Population

Suppose that DO with a population size of $I$ is employed to solve an optimization problem with the dimensional size of $D$. Each of the $i$-th dandelion seeds denoted as $X_i^t = \left[ x_{i,1}^t, \ldots, x_{i,d}^t, \ldots, x_{i,D}^t \right]$ is assumed to be a potential solution to solve the problem, where $i = 1, \ldots, I$ is the population indices of the dandelion seeds; $d = 1, \ldots, D$ indicates the dimensional index of the optimization problem; $t = 0, \ldots, T^{max}$ refers to the iteration index of DO with a maximum iteration number of $T^{max}$. Let the the upper and lower boundary limits of the solution be represented as $UB = [ub_1, \ldots, ub_d, \ldots, ub_D]$ and

$LB = [lb_1, \ldots, lb_d, \ldots, lb_D]$, respectively. During the initialization process where $t = 0$, the position vector of every $i$-th dandelion can be randomly generated as follows:

$$X_i^0 = LB + r_1(UB - LB) \tag{3}$$

where $r_1 \in [0, 1]$ is a random number generated with uniform distribution.

After the initial positions $X_i^0$ of all dandelion seeds are initialized using Equation (3), an operator $f(\cdot)$ is introduced to evaluate the fitness value of each initial dandelion seed as $f(X_i^0)$ for $i = 1, \ldots, I$. For the minimization problem, the $i$-th dandelion seed with smaller $f(X_i^t)$ value is considered better and vice versa. The initial population of DO is then sorted from the best to worse based on the fitness values of all initial dandelion seeds. After the sorting process, the dandelion seed with a population index of $i = 1$ is considered as the elite dandelion seed $X_{elite}^t$ with a fitness value of $f(X_{elite}^t)$, where:

$$\begin{cases} X_{elite}^t = X_1^t \\ f(X_{elite}^t) = f(X_1^t) \end{cases} \tag{4}$$

4.1.2. Rising Stage of DO

Each of the $i$-th dandelion seeds attempt to float away from its parent dandelion during the rising stage of DO, and its rising height is influenced by different weather factors such as humidity and wind speed. There are two types of weather considered in DO to simulate the flying ability of dandelion seeds during the rising stage.

A clear day is first type of weather considered in the rising stage to devise a search operator with more explorative behavior. Under this weather, the wind speed is modeled as a lognormal distribution that can generate more random numbers along its $y$-axis to increase the likelihood of dandelion seeds to be scattered in farther locations. For a lognormal distribution $\ln Y \sim N(\mu, \sigma^2)$ with mean and standard deviation values set as $\mu = 0$ and $\sigma = 1$, respectively, it is defined as:

$$\ln Y = \begin{cases} \frac{1}{y\sqrt{2\pi}} \exp\left[-\frac{1}{2\sigma^2}(\ln y)^2\right], & \text{if } y \geq 0 \\ 0 & , \quad \text{otherwise} \end{cases} \tag{5}$$

where $y$ is a random number generated from the standard normal distribution of $N(0, 1)$. It is noteworthy that the vortexes above the dandelion seeds can be adjusted based on the current wind speed and rise in spiral form. Particularly, the dandelion seeds can raise to higher heights and be propagated to farther regions with stronger wind and vice versa. Let $X_{rand}^t$ be a position vector randomly generated in solution space during the $t$-th iteration as:

$$X_{rand}^t = LB + rand(1, D) \times (UB - LB) \tag{6}$$

where $rand(1, D)$ is a vector consisting of $1 \times D$ real-valued numbers randomly generated from uniform distribution. For any $(t + 1)$-th iteration with a clear day, the new position $X_i^{t+1}$ of each $i$-th dandelion seed can be updated as follows:

$$X_i^{t+1} = X_i^t + \alpha v_x v_y \ln Y (X_{rand}^t - X_i^t) \tag{7}$$

Parameter $\alpha$ is a random perturbation between $[0, 1]$ calculated using Equation (8), and it is used to adjust the step size of the search process adaptively. Similar with $r_1$, $r_2 \in [0, 1]$ is a random number generated with uniform distribution. According to Equation (8), $\alpha$ is set as a relatively large value during the initial stage to promote exploration search, and it is nonlinearly decreased with the iteration numbers to emphasize the exploitation search in a later stage.

$$\alpha = r_2 \times \left[\frac{1}{(T^{max})^2}t^2 - \frac{2}{T^{max}}t + 1\right] \tag{8}$$

Let $\theta$ be a random number of ranges between $[-\pi, \pi]$ used to determine an amplitude value of $r$. The lift component coefficients applied on the $x$-axis and $y$-axis of a dandelion seed due to the eddy action are then calculated as $v_x$ and $v_y$, respectively, where:

$$\begin{cases} r = \frac{1}{\exp \theta} \\ v_x = r \times \cos \theta \\ v_y = r \times \sin \theta \end{cases} \tag{9}$$

Apart from a clear day, a rainy day is the second type of weather considered in the raising stage of DO to design a search operator with more exploitative behavior. Dandelion seeds are typically scattered around the neighborhood regions of the parent dandelion during a rainy day because they are unable to fly higher in the environments with stronger air resistance and higher humidity level. For any $(t + 1)$-th iteration with a rainy day, the new position $X_i^{t+1}$ of each $i$-th dandelion seed is updated as:

$$X_i^{t+1} = X_i^t \times k \tag{10}$$

where $k$ is a parameter with a downward convex oscillation pattern used to govern the neighborhood search range of a dandelion seed. Suppose that $q$ is parameter that gradually decreases from 1 to 0 with iteration number and $r_3 \in [0, 1]$ is random number generated from uniform distribution. Both $k$ and $q$ are then calculated as:

$$\begin{cases} q = \frac{1}{(T^{max})^2 - 2T^{max} + 1} t^2 - \frac{2}{(T^{max})^2 - 2T^{max} + 1} t + 1 + \frac{1}{(T^{max})^2 - 2T^{max} + 1} \\ k = 1 - r_3 \times q \end{cases} \tag{11}$$

To this end, the search operators used to update the new position $X_i^{t+1}$ of each $i$-th dandelion seed during the rising stage of DO are consolidated as:

$$X_i^{t+1} = \begin{cases} X_i^t + \alpha v_x v_y \ln Y \left( X_s^t - X_i^t \right), & \text{if } randn < 1.5 \\ X_i^t \times k, & \text{otherwise} \end{cases} \tag{12}$$

where $randn$ is a random number generated using standard normal distribution of $N(0, 1)$, and it is used to regulate the exploration and exploitation strengths of DO during the rising stage. A cutoff point of 1.5 is set in Equation (12) for DO to have stronger exploration strength in the first stage. This parameter setting enables all dandelion seeds to search within wider areas of solution space and provide more accurate directional information in the next iteration.

### 4.1.3. Descending Stage of DO

After the dandelion seeds have been raised to certain heights, they start to descend steadily. A search operator with explorative behavior is used to simulate the movement of dandelion seeds during the descending stage. Specifically, the moving trajectory of each dandelion seed in the descending stage can be updated iteratively based on Brownian motion. The directional information of population mean is leveraged to ensure that all dandelion seeds can descend steadily, enabling the population to search toward better solution regions of space. Suppose that $X_{mean}^t$ represents the mean position of the population in the $t$-th iteration, where:

$$X_{mean}^t = \frac{1}{I} \sum_{i=1}^{I} X_i^t \tag{13}$$

Define $\beta^t$ as a random number generated from the standard normal distribution of $N(0, 1)$, and it is used to emulate the Brownian motion [48]. Referring to $X_{mean}^t$ and $\beta^t$, the new position $X_i^{t+1}$ of each $i$-th dandelion seed can be updated in descending stage as:

$$X_i^{t+1} = X_i^t - \alpha \beta^t \left( X_{mean}^t - \alpha \beta^t X_i^t \right) \tag{14}$$

According to Equation (14), the directional information offered by population mean $X_{mean}^t$ has a crucial role to guide the search process of each dandelion seed in the descending stage. With the presence of Brownian motion, the dandelion seeds can perform searching with irregular motion during the descending stage, and the exploration strength offered in Equation (14) enables all population members to have greater chances in escaping from the local optima, hence reducing the likelihood of premature convergence.

### 4.1.4. Landing Stage of DO

After the rising and descending stages, a search operator with exploitative behavior is designed for the landing stage to simulate the behavior of dandelion seeds in choosing the random locations for landing. As the iteration number increases, all dandelion seeds of DO are expected to gradually converge toward global optimum by exploiting the neighborhood regions of promising solutions found.

A Levy flight coefficient denoted as $Levy(\lambda)$ is considered to simulate the step size of dandelion seeds during the landing stage of DO. The Levy flight function is chosen for the step size adjustment because it can encourage more local search domains with limited iterations numbers by allowing the dandelion seeds to stride to other positions with greater probability under Gaussian distribution [49], i.e.,

$$Levy(\lambda) = s \times \frac{w \times \sigma}{|v|^{\frac{1}{\widetilde{\beta}}}} \tag{15}$$

where $s$ is a fixed constant of 0.01; $\widetilde{\beta}$ is a parameter set as 1.5; and $w, v \in [0, 1]$ are two random numbers generated using normal distribution. Define $\Gamma(\cdot)$ as an operator used to represent Gamma distribution; then, $\sigma$ is calculated as:

$$\sigma = \left( \frac{\Gamma\left(1 + \widetilde{\beta}\right) \times \sin\left(\frac{\pi\widetilde{\beta}}{2}\right)}{\Gamma\left(\frac{1+\widetilde{\beta}}{2}\right) \times \widetilde{\beta} \times 2^{\left(\frac{\widetilde{\beta}-1}{2}\right)}} \right) \tag{16}$$

Define $\delta \in [0, 2]$ as a function that can linearly increase with time to avoid the excessive exploitation search of each dandelion seed, i.e.,

$$\delta = \frac{2t}{T^{max}} \tag{17}$$

Then, each dandelion seed $X_i^t$ can update its position in the landing stage based on the promising directional information of elite solution $X_{elite}^t$ to accelerate its convergence toward the global optimal solution as:

$$X_i^{t+1} = X_{elite}^t - Levy(\lambda) \times \alpha \times \left( X_{elite}^t - X_i^t \times \delta \right) \tag{18}$$

### 4.1.5. Overall Framework of DO

The pseudocode used to describe the overall search mechanisms of DO is presented in Algorithm 1. In the current study, three decision variables known as spindle speed, feed rate and point angle are to be optimized; hence, each dandelion seed is encoded as a position vector with $D = 3$. Meanwhile, the delamination sizes on the top and bottom sides of samples are the objective functions to be minimized.

At the beginning of search process, the initial positions of all dandelion seeds are randomly produced, and their respective fitness values are evaluated using the predefined objective functions. These dandelion seeds are then sorted from the best to worst based on their fitness values, where the first sorted dandelion seed is identified as an elite solution with the best performance for that particular iteration. During the iterative search process, the positions of all dandelion seeds are updated based on search operators inspired by the

rising, descending and landing stages to generate the new population. Similarly, the fitness values of all updated dandelion seeds in the new population are evaluated and sorted from best to worst before proceeding to the next iteration. This sorting process is beneficial to preserve the promising directional information of good performing dandelion seeds in the next iteration.

The iterative search processes of all dandelion seeds in three stages of rising, descending and landing of DO are repeated until the maximum iteration number is exceeded, i.e., $t > T^{max}$. At the end of the search process, the decision variables encoded in the best dandelion seed found so far (i.e., $X_{best}$) are returned as the optimal machining parameters used to minimize the predefined objective function. Since the inception of DO, it has been used to solve the CEC 2017 unconstrained benchmark functions [50] and four engineering design problems related to speed reducer, tension/compression spring, welded beam and pressure vessel. To be best of the authors' knowledge, this is the first study that attempts to apply DO for searching for the best combinations of machining parameters for drilling optimization problem.

---

**Algorithm 1**: DO

---

**Inputs:** $N$, $D$, $T^{max}$, $s$, $\widetilde{\beta}$, $UB$ and $LB$
01 : Initialize $t \leftarrow 0$ , $X_{best} \leftarrow \varnothing$ , $f(X_{best}) \leftarrow \infty$;
02:     **for** each $i$-th dandelion seed **do** /*Population Initialization*/
03 :       Randomly generate $X_i^0$ using Equation (3) and evaluate its fitness as $f\left(X_i^0\right)$;
04:     **end for**
05:     Sort all dandelion seeds from best to worst based on their fitness;
06 :     Update $X_{elite}^t$ and $f\left(X_{elite}^t\right)$ based on the sorted population using Equation (4);
07 :     $X_{best} \leftarrow X_{elite}^t$ , $f(X_{best}) \leftarrow f\left(X_{elite}^t\right)$;
08 :     $t \leftarrow t+1$;
09 :     **while** $t \leq T^{max}$ **do** /*Main Loop*/
10 :       **for** each $i$-th dandelion seed **do**
11 :           Randomly generate *randn* using $N(0,1)$;
12 :           **if** *randn* $< 1.5$ **then** /*Rising Stage*/
13 :               Calculate $\ln Y$, $X_{rand}^t$ using Equations (5), (6) and (8), respectively;
14:               Calculate, $v_x$ and $v_y$ using Equation (9);
15 :               Update $X_i^{t+1}$ using Equation (7); /*Clear Day*/
16:           **else**
17:               Calculate and $k$ using Equation (11);
18 :               Update Update $X_i^{t+1}$ using Equation (10); /*Raining Day*/
19 :           **end if**
20 :           Calculate $X_{mean}^t$ using Equation (13) and generate $\beta^t$ using $N(0,1)$;
21 :           Update $X_i^{t+1}$ using Equation (14); /*Descending Stage*/
22:           Calculate, $Levy(\lambda)$ and $\delta$ using Equations (16), (15) and (17), respectively.;
23 :           Update $X_i^{t+1}$ using Equation (18); /*Landing Stage*/
24 :           Fitness evaluation of $X_i^{t+1}$ as $f\left(X_i^{t+1}\right)$;
25 :       **end for**
26 :       Sort all updated dandelion seeds from best to worst based on their new fitness;
27 :       Update $X_{elite}^{t+1}$ and $f\left(X_{elite}^{t+1}\right)$ based on the sorted population using Equation (4);
28 :     **if** $f\left(X_{elite}^{t+1}\right)$ is better than $f(X_{best})$ **then**
29 :         $X_{best} \leftarrow X_{elite}^{t+1}$ , $f(X_{best}) \leftarrow f\left(X_{elite}^{t+1}\right)$;
30:     **end if**
31:     $t \leftarrow t+1$;
32:   **end while**
**Outputs:** $X_{best}$ and $f(X_{best})$

---

*4.2. Performance Evaluation of DO on Proposed Drilling Optimization Problems*

4.2.1. Simulation Settings

The performance of DO to solve the proposed drilling optimization problems is compared with seven other nature-inspired algorithms, i.e., PSO (particle swarm optimization) [51], DE (differential evolution) [52], TLBO (teaching-learning based optimization) [53], GWO (gray wolf optimizer) [54], SCA (sine cosine algorithm) [55], AOA (arithmetic optimization algorithm) [56] and GTO (gorilla troop optimizer) [57]. The optimal algorithmic-specific parameters of these nature-inspired algorithms are set according to the literature, and their original source codes available online are employed for fair comparison. All nature-inspired algorithms are simulated with the parameters of $I = 10$ and $T^{max} = 100$ to solve the proposed drilling optimization problem for 20 successive runs in order to address the random discrepancy issue. Finally, all compared algorithms are also implemented with Matlab (R2021a) installed on a personal desktop with an Intel® Core™ i5-7400 CPU processor with 24.0 GB RAM during performance analyses.

4.2.2. Simulation Results

Table 3 presents the machining parameters optimized by DO to minimize the delamination sizes at both the top and bottom sides. Given these optimal machining parameters, the differences between the predicted and actual delamination sizes at both the top and bottom sides are also compared for performance validation. According to Table 3, DO has produced the smallest predicted delamination size of 1.1259 at the top side when the machining parameters of spindle speed, feed rate and point angle are optimized as 2400 rpm, 0.1 mm/min, and 120°, respectively. Meanwhile, the optimal spindle speed of 2400 rpm, feed rate of 0.3 mm/min and point angle of 89° can result in the minimum predicted delamination of 0.8644 at the bottom side. Further experiments were conducted to validate these two sets of optimized machining parameters by measuring delamination at the top and bottom sides. On comparing the prediction results and validation experimental results as shown in Table 3, it can be concluded that there is good consistency between the simulation and experimental results due to their marginal deviations. Indeed, in actual experimentation, we can use either one of these settings. For instance, we postulate that bottom delamination can be avoided by using support plates beneath the sample, while drilling parameters can be optimized for minimal top delamination.

**Table 3.** Comparison between the predicted and actual delamination sizes at both the top and bottom sizes based on the optimal machining parameters found by DO.

| Side | Spindle Speed (rpm) | Feed Rate (mm/min) | Point Angle (°) | $F_d$ (Predicted) | $F_d$ (Actual) |
|---|---|---|---|---|---|
| Top | 2400 | 0.1 | 120 | 1.1259 | 1.1 |
| Bottom | 2400 | 0.3 | 89 | 0.8644 | 0.86 |

4.2.3. Verification and Comparison of Performance

The best, worst, mean and standard deviation (SD) values of delamination sizes produced by all eight nature-inspired algorithms at the top and bottom sides when solving the proposed drilling optimization problems for 20 successive runs are summarized in Tables 4 and 5, respectively. Bold fonts are used to indicate the best optimization results produced by the compared algorithms. Although the majority of the compared nature-inspired algorithms are able to produce the best (i.e., smallest) delamination sizes for the top and bottom sides except for GTO, DO is the only algorithm that can solve the proposed drilling optimization problems with the best results in terms of the worst and mean delamination sizes for both top and bottom sides, as reported in Tables 4 and 5, respectively. The excellent consistency of DO to solve the proposed drilling optimization problems with promising results can also be indicated from the small SD values. Other nature-inspired algorithms have demonstrated different behaviors when optimizing the

delamination sizes for the top and bottom sides. For instance, PSO and AOA tend to deliver better results when optimizing the delamination sizes at the bottom side, whereas GTO has demonstrated opposite behaviors and performed better in minimizing delamination sizes at the top side. The remaining nature-inspired algorithms (i.e., DE, TLBO, GWO and SCA) have shown the similar performance trends when optimizing the delamination sizes at the top and bottom sides.

**Table 4.** Comparisons of nature-inspired algorithms in terms of delamination sizes at the top side.

| Algorithm | $F_d$ (Top Side) | | | |
|---|---|---|---|---|
| | Best | Worst | Mean | Std Dev |
| PSO | 1.1259 | 1.1372 | 1.1290 | $3.303 \times 10^{-3}$ |
| DE | 1.1259 | 1.1378 | 1.1296 | $3.997 \times 10^{-3}$ |
| TLBO | 1.1259 | 1.1320 | 1.1265 | $1.881 \times 10^{-3}$ |
| GWO | 1.1259 | 1.1429 | 1.1284 | $4.874 \times 10^{-3}$ |
| SCA | 1.1259 | 1.1276 | 1.1262 | $4.944 \times 10^{-3}$ |
| AOA | 1.1259 | 1.1320 | 1.1278 | $2.873 \times 10^{-3}$ |
| GTO | 1.1259 | 1.1329 | 1.1272 | $2.661 \times 10^{-3}$ |
| DO | **1.1259** | **1.1259** | **1.1259** | $\mathbf{2.471 \times 10^{-11}}$ |

**Table 5.** Comparisons of nature-inspired algorithms in terms of delamination sizes at the bottom side.

| Algorithm | $F_d$ (Bottom Side) | | | |
|---|---|---|---|---|
| | Best | Worst | Mean | Std Dev |
| PSO | 0.8644 | 0.8815 | 0.86823 | $4.773 \times 10^{-3}$ |
| DE | 0.8644 | 0.9312 | 0.87683 | $1.888 \times 10^{-2}$ |
| TLBO | 0.8644 | 0.8824 | 0.86714 | $6.571 \times 10^{-3}$ |
| GWO | 0.8644 | 0.9225 | 0.87271 | $1.459 \times 10^{-2}$ |
| SCA | 0.8644 | 0.8911 | 0.86640 | $5.867 \times 10^{-3}$ |
| AOA | 0.8644 | 0.8670 | 0.86574 | $1.310 \times 10^{-3}$ |
| GTO | 0.8651 | 0.8824 | 0.87076 | $6.030 \times 10^{-3}$ |
| DO | **0.8644** | **0.8644** | **0.86440** | $\mathbf{2.833 \times 10^{-7}}$ |

The boxplots produced by all nature-inspired algorithms when solving the proposed drilling optimization problem with minimum delamination sizes at the top and bottoms sides are presented in Figure 8 to visualize the distributions of results in 20 simulation runs. Note that the symbol "+" appearing on the outside of the upper edge or below the lower edge of the boxplot implies the presence of extreme results during simulation. Referring to Figure 8A,B, it is observed that DO can produce much better optimization results than the other nature-inspired algorithms when minimizing the delamination sizes at both the top and bottom sides. The stability and robustness of DO are verified by the marginal deviation between the amplitudes of the upper and lower limits of its boxplots. Other nature-inspired algorithms such as PSO, DE, GWO, AOA and GTO have shown some inconsistencies in minimizing at least one of the delamination sizes, as revealed by large amplitudes between the upper and lower limits of their boxplots. Although the amplitudes of the upper and lower limits of the boxplots produced by TLBO are comparable with those of DO, some outliers can be observed from the boxplots of former algorithms. Based on the boxplot analyses, the effectiveness and robustness of DO in solving the proposed drilling optimization problems are verified.

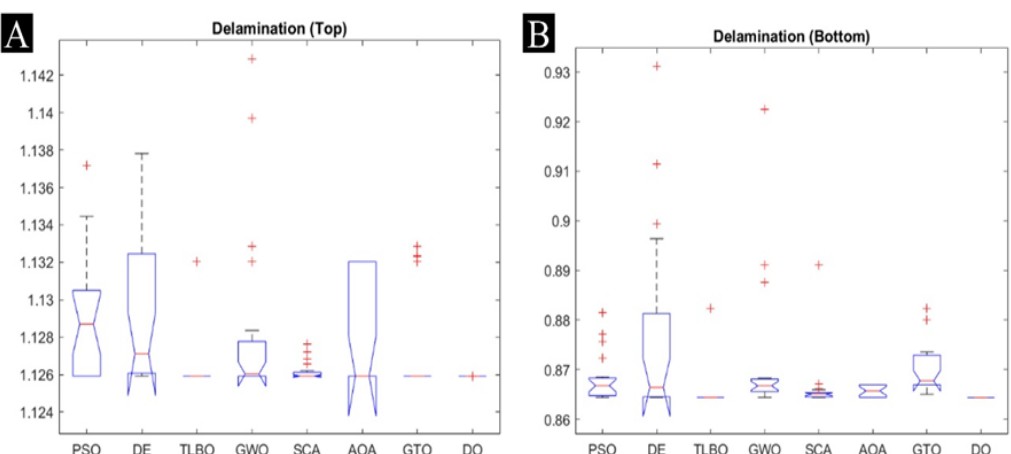

**Figure 8.** Boxplots of different nature-inspired algorithms when optimizing the delamination sizes at the (**A**) top and (**B**) bottom sides.

The convergence curves produced by all compared nature-inspired algorithms in minimizing the delamination sizes at the top and bottom sides are illustrated in Figure 9A,B, respectively. Both the GWO and DE are observed to solve the proposed drilling optimization problems with unsatisfactory delamination sizes at the top and bottom sides due to their slow convergence speeds in locating the promising solution regions. Although both of PSO and AOA are able to minimize the delamination size at the bottom side with relatively good accuracy, these two nature-inspired algorithms perform poorly in terms of solution accuracy and convergence speed when optimizing the delamination size at the top side. Meanwhile, TLBO, SCA and GTO have shown a high tendency to be trapped in local optima and suffer from a premature convergence issue despite having relatively promising convergence speeds in optimizing the delamination sizes at both the top and bottom sides. It is noteworthy that certain nature-inspired algorithms such as SCA, TLBO and SCA fail to search for the optimal machining parameters that can lead to the minimum delamination sizes at the top and bottom sides in all 20 simulation runs despite having better initial fitness values during the initial stage. This undesirable scenario can be justified by the fact that the best solutions of these nature-inspired algorithms were initialized in the local and have misled the remaining population members to converge around these inferior regions of solution space in certain simulation runs. In contrary to its competitors, DO is reported as the best-performing algorithm to solve the proposed drilling optimization problems by producing the smallest delamination sizes at the top and bottom sides. Despite being initialized with relatively large fitness values, Figure 9A,B verified the excellent convergence characteristic of DO because it has successfully searched for the optimal machining parameters that can lead to the minimum delaminating sizes at the top and bottom sides within 15 iterations for all 20 simulation runs. The four search operators formulated in the rising, descending and landing stages of DO are proven able to achieve better balancing of exploration and exploitation searches; hence, they can guide the population to locate the global optima of the proposed drilling optimization problem with fewer iteration numbers.

The performance comparison of all nature-inspired algorithms in solving the proposed drilling optimization problem is further investigated using non-parametric statistical analyses [58]. A pairwise comparison between DO and each nature-inspired algorithm with the Wilcoxon signed-rank test is first performed at the significance level of $\varphi = 0.05$. Tables 6 and 7 present the Wilcoxon signed-rank test results for pairwise comparison between DO and each nature-inspired algorithm when optimizing the delamination size at the top and bottom sides, respectively. Particularly, $R^+$ and $R^-$ represent the sum of rank for DO to outperform and underperform a given nature-inspired algorithm, respectively. The minimum level of significance used to identify the performance deviation between algorithms is indicated with the *p*-value, where the better performing algorithm is considered

to significantly outperform its competitors if $p < \sigma$. Referring to Tables 6 and 7, DO is reported to significantly outperform other nature-inspired algorithms because the $p$-value produced in each pairwise comparison is smaller than the threshold significance value of $\varphi = 0.05$. In other words, the outperformance of DO against other nature-inspired algorithms when solving the proposed drilling optimization problems is evident and not achieved by random chances.

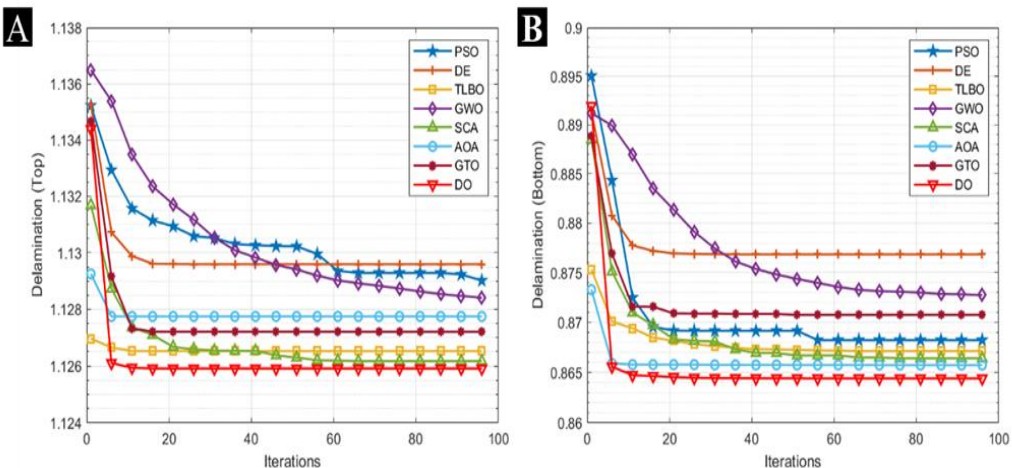

**Figure 9.** Convergence curves of different nature-inspired algorithms when optimizing the delamination sizes at the: (**A**) top and (**B**) bottom sides.

**Table 6.** Wilcoxon signed-rank test for the pairwise comparison between DO and other nature-inspired algorithms when optimizing the delamination size at the top side.

| DO vs. | $R^+$ | $R^-$ | $p$-Value |
|---|---|---|---|
| PSO | 210 | 0 | $4.30 \times 10^{-5}$ |
| DE | 210 | 0 | $4.30 \times 10^{-5}$ |
| TLBO | 210 | 0 | $4.30 \times 10^{-5}$ |
| GWO | 210 | 0 | $4.30 \times 10^{-5}$ |
| SCA | 210 | 0 | $4.30 \times 10^{-5}$ |
| AOA | 210 | 0 | $4.30 \times 10^{-5}$ |
| GTO | 210 | 0 | $4.30 \times 10^{-5}$ |

**Table 7.** Wilcoxon signed-rank test for the pairwise comparison between DO and other nature-inspired algorithms when optimizing the delamination size at the bottom side.

| DO vs. | $R^+$ | $R^-$ | $p$-Value |
|---|---|---|---|
| PSO | 210 | 0 | $8.20 \times 10^{-5}$ |
| DE | 210 | 0 | $8.20 \times 10^{-5}$ |
| TLBO | 210 | 0 | $8.20 \times 10^{-5}$ |
| GWO | 210 | 0 | $8.20 \times 10^{-5}$ |
| SCA | 210 | 0 | $8.20 \times 10^{-5}$ |
| AOA | 210 | 0 | $8.20 \times 10^{-5}$ |
| GTO | 210 | 0 | $8.20 \times 10^{-5}$ |

Multiple comparison analyses are subsequently performed to investigate the performance differences between selected nature-inspired algorithms more thoroughly [58]. The Friedman test is used to calculate the average ranking of each nature-inspired algorithm and the $p$-value that has a crucial role in identifying the global differences between these nature-inspired algorithms. Table 8 reveals that DO has achieved the best (i.e., lowest) rank value in minimizing the delamination size at the top side, which is followed by TLBO,

GTO, SCA, AOA, GWO, PSO and DE. Similarly, DO also performs the best when optimizing the delamination size at the bottom side according to Table 9, which is followed by TLBO, PSO, DE and the remaining four nature-inspired algorithms (i.e., GWO, SCA, AOA and GTO) with the same rank values. Notably, the *p*-values presented in both Tables 8 and 9 are smaller than $\varphi = 0.05$, implying the presence of global differences between all eight compared nature-inspired algorithms. Another three post hoc procedures, i.e., Bonferroni–Dunn, Holm and Hochberg methods, are subsequently employed to detect concrete differences between DO and each nature-inspired algorithm, and their results in terms of *z* values, unadjusted *p*-value and adjusted *p*-value (APV) are presented in Tables 10 and 11. Particularly, Table 10 reports that all post hoc methods have confirmed the superiority of DO against other nature-inspired algorithms when optimizing the delamination size at the top side because all the APVs produced are smaller than $\varphi = 0.05$. For Table 11, it reveals that DO can significantly outperform DE, PSO, GWO, AOA, SCA and GTO in minimizing the delamination size at the bottom side, as confirmed by all three post hoc methods. Meanwhile, only Holm and Hochberg can detect the significant performance of DO against TLBO when minimizing the delamination size at the bottom side.

**Table 8.** Friedman test for multiple comparisons between DO and other nature-inspired algorithms when optimizing the delamination size at the top side.

| Nature-Inspired Algorithm | Ranking | Chi-Square Statistic | *p*-Value |
|---|---|---|---|
| PSO | 5.875 | | |
| DE | 6.425 | | |
| TLBO | 3.950 | | |
| GWO | 5.275 | | |
| SCA | 4.425 | 62.670833 | $0.00 \times 10^0$ |
| AOA | 4.675 | | |
| GTO | 4.375 | | |
| DO | 1.000 | | |

**Table 9.** Friedman test for multiple comparisons between DO and other nature-inspired algorithms when optimizing the delamination size at the bottom side.

| Nature-Inspired Algorithm | Ranking | Chi-Square Statistic | *p*-Value |
|---|---|---|---|
| PSO | 4.925 | | |
| DE | 5.275 | | |
| TLBO | 3.000 | | |
| GWO | 5.450 | | |
| SCA | 5.450 | 62.970833 | $0.00 \times 10^0$ |
| AOA | 5.450 | | |
| GTO | 5.450 | | |
| DO | 1.000 | | |

**Table 10.** Post hoc analyses for the multiple comparisons between DO and other nature-inspired algorithms when optimizing the delamination size at the top side.

| DO vs. | Unadjusted *p* | Bonferroni–Dunn *p* | Holm *p* | Hochberg *p* |
|---|---|---|---|---|
| PSO | $0.00 \times 10^0$ | $0.00 \times 10^0$ | $0.00 \times 10^0$ | $0.00 \times 10^0$ |
| DE | $0.00 \times 10^0$ | $0.00 \times 10^0$ | $0.00 \times 10^0$ | $0.00 \times 10^0$ |
| TLBO | $1.40 \times 10^{-4}$ | $9.79 \times 10^{-4}$ | $1.40 \times 10^{-4}$ | $1.40 \times 10^{-4}$ |
| GWO | $0.00 \times 10^0$ | $0.00 \times 10^0$ | $0.00 \times 10^0$ | $0.00 \times 10^0$ |
| SCA | $1.00 \times 10^{-5}$ | $6.90 \times 10^{-5}$ | $2.90 \times 10^{-5}$ | $2.60 \times 10^{-5}$ |
| AOA | $2.00 \times 10^{-6}$ | $1.50 \times 10^{-5}$ | $8.00 \times 10^{-6}$ | $8.00 \times 10^{-6}$ |
| GTO | $1.30 \times 10^{-5}$ | $9.20 \times 10^{-5}$ | $2.90 \times 10^{-5}$ | $2.60 \times 10^{-5}$ |

**Table 11.** Post hoc analyses for the multiple comparisons between DO and other nature-inspired algorithms when optimizing the delamination size at the bottom side.

| DO vs. | Unadjusted $p$ | Bonferroni–Dunn $p$ | Holm $p$ | Hochberg $p$ |
|---|---|---|---|---|
| PSO | $0.00 \times 10^0$ | $3.00 \times 10^{-6}$ | $1.00 \times 10^{-6}$ | $1.00 \times 10^{-6}$ |
| DE | $0.00 \times 10^0$ | $0.00 \times 10^0$ | $0.00 \times 10^0$ | $0.00 \times 10^0$ |
| TLBO | $9.82 \times 10^{-3}$ | $6.88 \times 10^{-2}$ | $9.82 \times 10^{-3}$ | $9.82 \times 10^{-3}$ |
| GWO | $0.00 \times 10^0$ | $0.00 \times 10^0$ | $0.00 \times 10^0$ | $0.00 \times 10^0$ |
| SCA | $0.00 \times 10^0$ | $0.00 \times 10^0$ | $0.00 \times 10^0$ | $0.00 \times 10^0$ |
| AOA | $0.00 \times 10^0$ | $0.00 \times 10^0$ | $0.00 \times 10^0$ | $0.00 \times 10^0$ |
| GTO | $0.00 \times 10^0$ | $0.00 \times 10^0$ | $0.00 \times 10^0$ | $0.00 \times 10^0$ |

## 5. Conclusions

In conclusion, the present study successfully investigated the mechanical properties of hybrid carbon and glass fiber-reinforced composite laminates and the effect of drilling parameters on the delamination factor and failure mechanism of the composites. Experimental findings from drilling and delamination analysis show that increasing the point angle from 80° to 120° resulted in low delamination upon entry, a spindle speed of 1000 rpm resulted in lower delamination at entry compared to higher spindle speeds, while a feed rate of 0.2 mm/min resulted in lower delamination at entry compared to feed rates of 0.1 to 0.3 mm/min. The optimization findings revealed that an optimal spindle speed of 2400 rpm, feed rate of 0.3 mm/min and point angle of 89° result in a minimal predicted delamination of 0.8644 (bottom side), which consistently agreed with the experimental findings. It was ascertained that DO produced much better optimization results compared to other nature-inspired algorithms when minimizing delamination sizes due to the stability and robustness of DO (i.e., marginal deviation between the amplitudes of the upper and lower limits of boxplots). Future studies will focus on the effects of the stacking sequence of the hybrid fibers on machinability, delamination-induced damages as well the failure mode (i.e., brittle or ductile) of the laminates.

**Author Contributions:** Conceptualization, E.N., K.M. and G.F.; methodology, S.N., S.M.S. and A.D.A.S.; software, K.V. and W.H.L.; validation, E.N., K.M. and G.F.; formal analysis, E.N.; investigation, S.M.S., A.D.A.S. and S.N.; resources, S.M.S., S.N. and A.D.A.S., data curation, S.N.; writing—original draft preparation, K.M.; writing—review and editing, K.M., W.H.L. and E.N.; visualization, S.N.; supervision, E.N.; project administration, E.N.; funding acquisition, G.F. All authors have read and agreed to the published version of the manuscript.

**Funding:** This research received no external funding.

**Institutional Review Board Statement:** Not applicable.

**Informed Consent Statement:** Not applicable.

**Data Availability Statement:** Not applicable.

**Conflicts of Interest:** The authors declare no conflict of interest.

## Nomenclature

**Abbreviations**

| | |
|---|---|
| DO | Dandelion optimizer |
| MSA | Metaheuristic search algorithm |
| CFRP | Carbon fiber-reinforced polymer |
| CF | Carbon fiber |
| GF | Glass fiber |
| SEM | Scanning electron microscopy |
| XRD | X-ray diffraction |
| PA | Polyamide |
| UCFF | Uncut fiber factor |

| DoE | Design of experiments |
|---|---|
| PSO | Particle swarm optimization |
| DE | Differential evolution |
| TLBO | Teaching learning-based optimization |
| GWO | Gray wolf optimizer |
| SCA | Sine cosine algorithm |
| AOA | Arithmetic optimization algorithm |
| GTO | Gorilla troops optimizer |
| **Indices** | |
| $i$ | Population index of dandelion seed in DO |
| $d$ | Dimension index |
| $t$ | Iteration index |
| **Operators** | |
| $f(\cdot)$ | Operator used to calculate the fitness value of a given position value |
| $N(\cdot, \cdot)$ | Operator used to calculate normal distribution |
| $Levy(\cdot)$ | Operator used to calculate the Levy flight function |
| $\Gamma(\cdot)$ | Operator used to calculate the Gamma distribution |
| **Parameter and Variables** | |
| $F_z$ | Thrust force |
| $F_d$ | Delamination factor |
| $D_{max}$ | Maximum delaminated diameter |
| $D_0$ | Nominal diameter of the hole |
| $A_0$ | Area between the hole circle and maximum delamination zone |
| $A_1$ | Area between the hole circle and the minimum damage zone |
| $A_H$ | Area of the drill |
| $D$ | Total dimension size of optimization problem |
| $I$ | Population size |
| $X_i^t$ | Position value of $i$-th dandelion seed at the $t$-th iteration |
| $T^{max}$ | Maximum iteration numbers |
| $UB$ | Upper boundary limits |
| $LB$ | Lower boundary limits |
| $X_i^0$ | Initial position value of $i$-th dandelion seed |
| $r_1, r_2, r_3,$ | Real-valued number within $[0, 1]$ randomly generated from uniform distribution |
| $rand(1, D)$ | Vector consists of $1 \times D$ real-valued numbers randomly generated from uniform distribution |
| $y, randn, w, v$ | Random number generated from standard normal distribution |
| $X_{elite}^t$ | Position value of elite dandelion seed at the $t$-th iteration |
| $\mu$ | Mean value of normal distribution |
| $\sigma$ | Standard deviation value of normal distribution |
| $\ln Y$ | Wind speed modeled using lognormal distribution |
| $X_{rand}^t$ | Position vector randomly generated in solution space during the $t$-th iteration |
| $\alpha$ | Random perturbation used to adjust the step size of search process adaptively |
| $r$ | Amplitude value of dandelion seed in rising stage of DO |
| $\theta$ | Random number of ranges between $[-\pi, \pi]$ |
| $v_x$ | Lift component coefficient applied on the $x$-axis of dandelion seed |
| $v_y$ | Lift component coefficient applied on the $y$-axis of dandelion seed |
| $k$ | Downward convex oscillation pattern of dandelion seed |
| $q$ | Parameter gradually decreases from 1 to 0 with iteration numbers |
| $X_{mean}^t$ | Mean position of population in the $t$-th iteration |
| $\beta^t$ | Random number generated from standard normal distribution and used to emulate Brownian motion |
| $s$ | Fixed constant with value of 0.01 |
| $\widetilde{\beta}$ | Parameter with value of 1.5 |
| $\delta$ | Parameter gradually increases from 0 to 2 with iteration numbers |
| $SD$ | Standard deviation |
| $\varphi$ | Significant level of non-parametric statistical procedures |
| $R^+$ | Sum of rank for DO to perform better than a given competing algorithm |
| $R^-$ | Sum of rank for DO to perform worse than a given competing algorithm |

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
