# Peer review of "Drilling-Induced Damages in Hybrid Carbon and Glass Fiber-Reinforced Composite Laminate and Optimized Drilling Parameters"

_jcs, doi:10.3390/jcs6100310_

Round 1
Reviewer 1 Report
The paper is focusing on an interesting topic however it is difficult to read, and some complementary explanations are highly needed in the revised version. The manuscript would benefit, once revised content-wise. The following points should be considered by the authors in the revised manuscript before acceptance for publication.
· The reviewer has a concern regarding the novelty of this paper. The authors should clarify the difference between the submitted paper and the following published paper:
1. Gemi, L., Köklü, U., Yazman, Åž., & Morkavuk, S. (2020). The effects of stacking sequence on drilling machinability of filament wound hybrid composite pipes: Part-1 mechanical characterization and drilling tests. Composites Part B: Engineering, 186, 107787.
2. Gemi, L., Morkavuk, S., Köklü, U., & Yazman, Åž. (2020). The effects of stacking sequence on drilling machinability of filament wound hybrid composite pipes: Part-2 damage analysis and surface quality. Composite Structures, 235, 111737.
• Some symbols don’t have description in the paper. It would better if a NOTATION added at the interval of the KEYWORDS and the INTRODUCTION, to present all the symbols in this paper.
• The introduction section has documented "well" the recent and relevant literature. However, the new contribution of this work is not clear. In other words, what is missing and has not been investigated in the previous literature which is addressed in this manuscript. Could the authors comment.
• The last paragraph of the introduction should be really improved. Indeed, the authors must rephrase this part in terms of experimental findings and to emphasis the paper’s structuration.
• It’s very important that the authors add more results about the effect of the frequency, the laser modes, thickness of specimens.
• The relation between the laser power vs scanning speed, and laser power vs toughness is not clear. I ask the authors to be constructive and give more details.
• It is a good report, but which are the lessons learnt? The authors have to clarify this otherwise the paper cannot be accepted
• The authors must improve the quality of all figures (300dpi)
Author Response
Thanks for your valuable comments.
All comments have been amended in the revised manuscript.

Reviewer 2 Report
The paper presents an interesting approach based on the Drilling Induced Damages in Hybrid Carbon and Glass Fibers Reinforced Composite Laminate and Optimized Drilling Parameters. However, the innovation of the current research work should be further highlighted and emphasized. At the same time, the authors should consider the following comments to greatly improve the quality of the paper.
1. In the abstract, introduce the problem in the initial lines of the abstract.
2. The introduction needs to be improved by relating to the mechanics of the studied materials and their mechanical characteristics. The references to be included are: 10.1177/0021998318790093, 10.1016/j.polymertesting.2017.09.009, 10.1016/j.compstruct.2021.114698, 10.1177/0731684417727143, 10.1002/app.46770, 10.1016/j.porgcoat.2022.107015.
3. Kindly add a table that describes the main physical and chemical properties of the raw materials used in this study.
4. Were the preparation methods described by the authors come in accordance with a certain standard or do they follow previous procedures?
5. The scale bar is not clear in the images of figure 2. Kindly enlarge the images and add a clear scale bar.
6. In Section 3.3, the author just reported few numbers that relate to the mechanical strength of the materials. There should be a link with the literature in this section. You need to compare these numbers against what other researchers had done.
7. The conclusion needs to be modified to summarize the research outcomes in short statements with clear observations.
Author Response

(The authors gave the same response as above.)

Round 2
Reviewer 1 Report
the Authors did not makes the major changes asked during the review process
Author Response
Dear Reviewer,
We have amended all your comments. Added more information in the updated manuscript.

Reviewer 2 Report
The authors are requested to kindly include these references in the introduction as mentioned in the original report.
10.1177/0021998318790093,
10.1016/j.polymertesting.2017.09.009,
10.1177/0731684417727143,
10.1016/j.porgcoat.2022.107015
Author Response
We have added relevant literatures.

Round 3
Reviewer 1 Report
The paper can be accepted in the present form.